# The leukemic oncogene *EVI1* hijacks a *MYC* super-enhancer by CTCF-facilitated loops

Sophie Ottema[1,2,7], Roger Mulet-Lazaro [1,2,7], Claudia Erpelinck-Verschueren [1,2,7], Stanley van Herk [1,2], Marije Havermans[1,2], Andrea Arricibita Varea[1,2], Michael Vermeulen[1], H. Berna Beverloo[3], Stefan Gröschel[4,5], Torsten Haferlach[6], Claudia Haferlach[6], Bas J. Wouters[1,2], Eric Bindels[1], Leonie Smeenk [1,2,8] & Ruud Delwel[1,2,8 ✉]

Chromosomal rearrangements are a frequent cause of oncogene deregulation in human malignancies. Overexpression of *EVI1* is found in a subgroup of acute myeloid leukemia (AML) with 3q26 chromosomal rearrangements, which is often therapy resistant. In AMLs harboring a t(3;8)(q26;q24), we observed the translocation of a *MYC* super-enhancer (*MYC* SE) to the *EVI1* locus. We generated an in vitro model mimicking a patient-based t(3;8) (q26;q24) using CRISPR-Cas9 technology and demonstrated hyperactivation of *EVI1* by the hijacked *MYC* SE. This *MYC* SE contains multiple enhancer modules, of which only one recruits transcription factors active in early hematopoiesis. This enhancer module is critical for *EVI1* overexpression as well as enhancer-promoter interaction. Multiple CTCF binding regions in the *MYC* SE facilitate this enhancer-promoter interaction, which also involves a CTCF binding site upstream of the *EVI1* promoter. We hypothesize that this CTCF site acts as an enhancer-docking site in t(3;8) AML. Genomic analyses of other 3q26-rearranged AML patient cells point to a common mechanism by which *EVI1* uses this docking site to hijack enhancers active in early hematopoiesis.

[1] Department of Hematology, Erasmus MC Cancer Institute, Rotterdam, The Netherlands. [2] Oncode Institute, Utrecht, The Netherlands. [3] Department of Clinical Genetics, Erasmus University Medical Center, Rotterdam, The Netherlands. [4] A380, German Cancer Research Center, Heidelberg, Germany. [5] Department of Internal Medicine V, Heidelberg University Hospital, Heidelberg, Germany. [6] Munich Leukemia Laboratory, Munich, Germany. [7] These authors contributed equally: Sophie Ottema, Roger Mulet-Lazaro, Claudia Erpelinck-Verschueren. [8] These authors jointly supervised this work: Leonie Smeenk, Ruud Delwel. ✉ email: h.delwel@erasmusmc.nl

The expression of cell lineage-specific genes is highly regulated. Specific enhancer-promoter interactions and transcription factor binding to regulatory elements delineate gene expression profiles that define cell identity and function[1]. Physical interactions between enhancers and promoters primarily occur within chromosome segments enclosed by chromatin loops known as topologically associated domains (TADs)[2]. TADs are separated from each other by boundaries typically containing convergent CTCF (CCCTC-binding factor) occupied sites[3]. According to the loop extrusion model, the cohesin complex catalyzes the formation of loops and CTCF dimers act as anchors to these loops[4]. CTCF and the cohesin complex, but also other factors like Ying Yang 1 (YY1), may also contribute to enhancer-promoter looping[5–8]. However, not all promoters or enhancers within a TAD interact with each other. The mechanisms by which promoters interact with certain enhancers and not with others are not fully understood[9,10]. Transcriptional control of genes driven by particular enhancer-promoter combinations depends on the availability of transcription factors and their ability to bind specific regulatory elements[8,11].

Chromosomal rearrangements frequently lead to changes in the expression or function of genes causing malignant transformation[12]. Often breakpoints are found within gene bodies, resulting in fusion oncogenes driving tumorigenesis[13]. Alternatively, when a regulatory element of a certain gene is translocated into the vicinity of another gene, it can lead to deregulation of both the donor and the acceptor genes. Well-described examples are the inv(3)(q21q26) or t(3;3)(q21;q26) rearrangements in acute myeloid leukemia (inv(3)/t(3;3) AML), in which a GATA2 enhancer at 3q21 is hijacked by EVI1 at 3q26, causing EVI1 overexpression and GATA2 haploinsufficiency[14,15]. AML is a heterogeneous disease, with EVI1 positive (EVI1+) inv(3)/t(3;3) patients being identified as a subgroup with a very poor response to therapy[16–19]. Besides inv(3)/t(3;3), many other EVI1 + AML cases with 3q26 rearrangements have been reported, including translocations t(2;3)(p21;q26), t(3;7)(q26;q24), t(3;6)(q26;q11), and t(3;8)(q26;q24)[18,20–27]. We hypothesize that in all these rearrangements EVI1 overexpression is induced by the repositioning of an enhancer that can interact with the EVI1 promoter, as shown for inv(3)/t(3;3) AML[14,15]. We performed targeted next-generation sequencing (NGS) of the long arm of chromosome 3 (3q-seq) in translocation t(3;8)(q26;q24) AML harboring an EVI1/MYC rearrangement[22,27]. Applying CRISPR-Cas9 technology, we generated a human t(3;8) cell line model with an eGFP reporter cloned 3' of EVI1. This unique model was used to investigate how enhancer-promoter interactions drive oncogenic EVI1 expression in leukemia. We demonstrate that CTCF in combination with transcription factors active in early hematopoiesis is essential in enhancer hijacking and oncogene activation.

## Results

**MYC super-enhancer translocation and EVI1 overexpression in t(3;8)(q26;q24) AML.** Using 3q-seq, the exact chromosomal breakpoints were determined in 10 AML samples with a translocation t(3;8)(q26;q24), hereafter referred to as t(3;8) AML. All breakpoints at 3q26.2 occurred upstream of the EVI1 promoter (Fig. 1a). At chromosome 8, the breakpoints were downstream of the oncogene MYC at 8q24, leaving the gene intact at its original location. In all 10 cases, a genomic region reported as a MYC super-enhancer (SE) had been translocated to EVI1 (Fig. 1b). The MYC SE harbors approximately 150 Kb of open chromatin enriched with histone mark H3K27 acetylation (H3K27ac) and is located 1.7 Mb downstream of MYC (Fig. 1b). This locus has been reported to be essential for transcriptional control of MYC

expression in normal hematopoiesis[28]. H3K27ac determined by ChIP-seq revealed EVI1 promoter activity in t(3;8) AML patient cells, comparable to the promoter activity in AML with inv(3)(q21q26). H3K27ac was absent at the EVI1 promoter in EVI1 negative (EVI1⁻) non-3q26 AML (Fig. 1a, lower panel). Accordingly, EVI1 expression was found to be highly elevated in t(3;8) compared to non-3q26 rearranged AMLs (Fig. 1c). The EVI1 levels in t(3;8) AMLs were comparable to the levels found in AMLs with inv(3)/t(3;3). These data support the hypothesis that EVI1 overexpression in t(3;8) AML is caused by the translocation of the MYC SE.

**A t(3;8) cell model recapitulates EVI1 overexpression in human AML.** To study the transcriptional activation of EVI1 by the MYC SE, we generated a human myeloid cell model with a translocation t(3;8)(q26;q24). We introduced eGFP in frame with a T2A self-cleavage site downstream of EVI1 in K562 cells (Fig. 2a). Successful integration of the insert is shown for two clones by flow cytometry and PCR (Fig. 2b, Supplementary Fig. 1a–c). Decreased eGFP levels were observed in the K562 EVI1-eGFP model after shRNA-directed EVI1 knockdown (Fig. 2c, d and Supplementary Fig. 1d–g). Next, sgRNAs for CRISPR-Cas9 editing were designed based on the genomic breakpoints of one of the t(3;8) AML patients in our cohort (Fig. 1a). Double strand DNA breaks were generated at 3q26 and 8q24 (Fig. 2e) using those guides. We hypothesized that the translocated MYC SE can activate EVI1 transcription, which consequently leads to increased eGFP levels. As shown in Fig. 2f, less than 0.1% of the sgRNA-treated K562 EVI1-eGFP cells showed increased eGFP levels. After two consecutive rounds of FACS sorting in combination with cell culture expansion, we obtained 95% eGFP positive cells of which single clones were isolated by single-cell sorting (process done similarly for both clones 8 and 24, Fig. 2f shows clone 24). The presence of a t(3;8) was demonstrated for four of these clones by PCR (Clone 24–7, Fig. 2h) and Sanger sequencing (Supplementary Fig. 2a). A combination of three separate diagnostic FISH probes for MECOM, MYC, and centromere chromosome 8 confirmed the successful generation of a translocation t(3;8) in all four clones (Supplementary Fig. 2b–e). The translocation caused a strong increase of mRNA and protein levels of EVI1, as well as of eGFP expression (Fig. 2c, g, j, k, l). No significant difference in MYC expression was observed between the parental K562 EVI1-eGFP and t(3;8) clones (Fig. 2i). Upon EVI1 knockdown by shRNA, eGFP and EVI1 expression were reduced as shown for clones 24-7 and 8-4 (Fig. 2l–m, and Supplementary Fig. 2f–g). We conclude that eGFP is a sensitive and reliable marker for EVI1 expression in this EVI1-eGFP t(3;8) model and that the translocated MYC SE strongly enhances EVI1 transcription.

**EVI1 promoter hyperactivation upon interaction with MYC SE in t(3;8) AML.** 4C-seq experiments taking the EVI1 promoter (EVI1_PR) as a viewpoint revealed specific interaction with the MYC SE in EVI1-eGFP t(3;8) cells, which was not found in the parental K562 EVI1-eGFP line (clone 24-7 and clone 24, respectively, Fig. 3a). This t(3;8)-specific interaction between the EVI1 promoter and MYC SE was confirmed in t(3;8) clone 8-4 (Supplementary Fig. 3d) and by reciprocal 4C-seq using the MYC SE as a viewpoint (clone 24-7, Fig. 3b). A comparable EVI1 promoter–MYC SE interaction was found in a primary t(3;8) AML sample (Fig. 3a, b), confirming that the K562 EVI1-eGFP t(3;8) model recapitulates primary AML. ChIP-seq for H3K4 trimethylation (H3K4me3, Fig. 3c) indicated the presence of an active EVI1 promoter in all K562 clones. However, H3K27 and H3K9 acetylation (H3K27ac and H3K9ac) levels were strongly

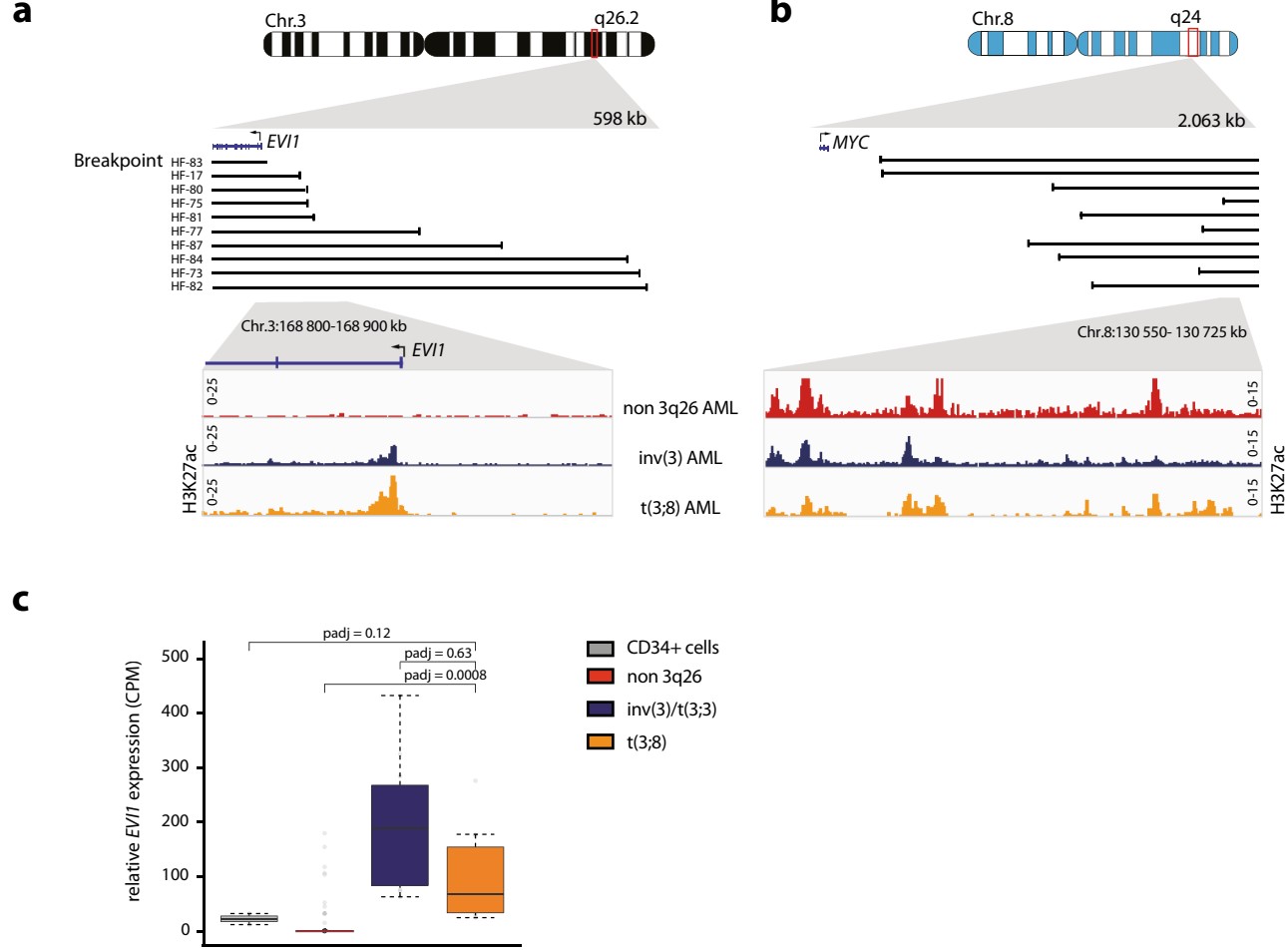

**Fig. 1 *MYC* super-enhancer translocation and *EVI1* overexpression in t(3;8)(q26;q24) AML. a** Upper part, schematic depiction of Chr.3, zoomed in on 3q26.2. Black lines correspond to sample-specific breakpoints detected by 3q-seq for each indicated t(3;8)(q26;q24) patient. Lower part: zoom-in on the *EVI1* promoter, H3K27ac ChIP-seq data for a primary non-3q26 AML sample in red (*N* = 1, AML-185), an inv(3)(q21q26) in blue (*N* = 1, AML-2190) and a t(3;8)(q26;q24) in orange (*N* = 1, AML-17). **b** Similar to A, but here in the upper part a schematic depiction of Chr.8, zoomed in on 8q24. Lower part: H3K27ac ChIP-seq data as in **a**, but here is a zoom-in on the +1.7 Mb *MYC* super-enhancer. **c** *EVI1* expression measured by RNA-seq in counts per million (CPM) for normal CD34 + HSPCs (*N* = 9, gray), non-3q26 AMLs (*N* = 114, red), inv(3)/t(3;3)(q21;q26) AMLs (*N* = 11, blue), and t(3;8)(q26;q24) AMLs (*N* = 10, orange). The lower and upper edges of the boxplots represent the first and third quartiles, respectively, the horizontal line inside the box indicates the median. The whiskers extend to the most extreme values within the range comprised between the median and 1.5 times the interquartile range. The circles represent outliers outside this range. The statistical significance of the comparisons between these groups was determined by the Wald test in the DESeq2 package. Adjusted *p*-values (padj) following multiple testing corrections by the Benjamini–Hochberg procedure are displayed.

increased at the promoter in all four t(3;8) clones, revealing a hyperactivated *EVI1* promoter (Fig. 3d, e) upon interaction with the translocated *MYC* SE.

**One critical enhancer module in the *MYC* SE drives *EVI1* transcription.** The *MYC* SE is a cluster of multiple individual enhancer modules that may recruit different sets of transcription factors[28]. To investigate which of the enhancer modules are driving oncogenic *EVI1* transcription in t(3;8) AML, we designed sgRNAs to sequentially delete those individual modules. H3K27ac ChIP-seq data of a primary t(3;8) AML and of t(3;8) clone 24-7 were used to illustrate the different enhancer modules A-I described previously[28] (Fig. 4a). The deletion of these modules by CRISPR-Cas9 using specific sgRNA pairs was shown by PCR and the effect on *EVI1* expression was determined by flow cytometry (Fig. 4b). Only the deletion of module C caused a loss of *EVI1/eGFP* expression. Due to the existence of multiple alleles (K562 has trisomy 8) and the partial efficiency of CRISPR-Cas9 in creating deletions, the translocated allele is exclusively targeted in

a subpopulation of cells. As a consequence, not all cells lose *EVI1* expression and show a GFP shift in the flow cytometry plot. A loss of *EVI1* mRNA and EVI1 protein levels was observed in the eGFP negative sorted cell fraction when module C was deleted (Fig. 4c–e and Supplementary Fig. 3a). In a control clone in which *EVI1-eGFP* expression was increased due to the amplification of *EVI1* instead of the translocation of the *MYC* SE (Supplementary Fig. 4a–e), the expression of *EVI1*-eGFP was not affected by mutating the *MYC* SE (Supplementary Fig. 4f). ATAC-seq and H3K27ac ChIP-seq in t(3;8) AML patients showed that module C was distinctly accessible and active compared to other modules (Supplementary Fig. 3b–c). Furthermore, ChIP-seq data revealed binding of early hematopoietic regulators (GATA2, FLI1, ERG, RUNX1, LMO2, and LYL1) to module C in CD34+ hemato-poietic stem and progenitor cells (HSPCs)[29] (Fig. 4f). Similar transcription factor binding patterns were found in t(3;8) AML patients and K562 cells, further confirming the functional significance of this module in this context (Supplementary Fig. 3c).

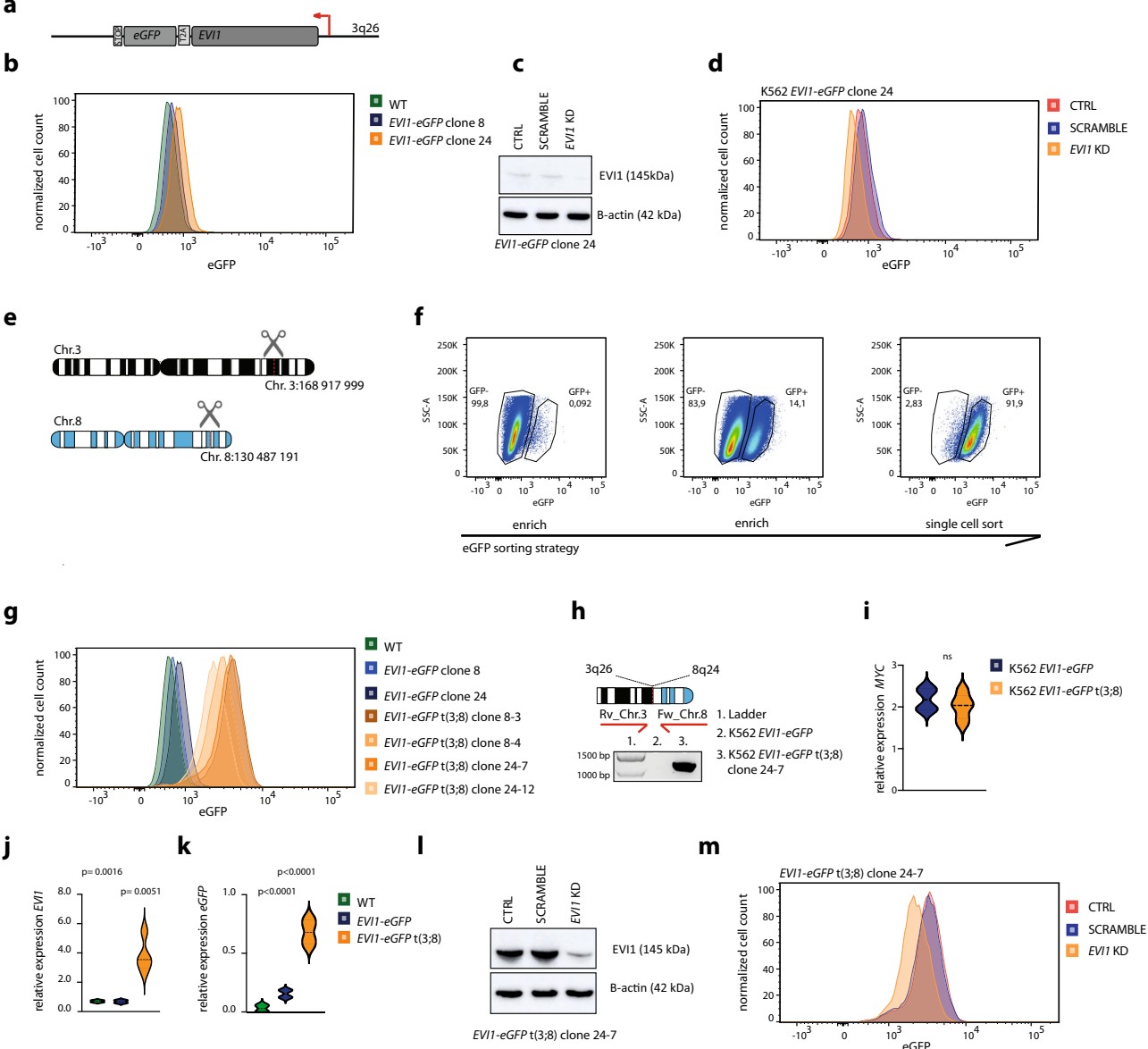

**Fig. 2 A t(3;8) cell model recapitulates *EVI1* overexpression in human AML. a** Schematic overview of *EVI1-T2A-eGFP*. **b** Flow cytometry plot presenting eGFP levels in K562-*EVI1-eGFP* clones. **c** Western blot shows EVI1 levels after shRNA-directed *EVI1* knockdown (KD), compared to the control and scrambled shRNA in K562 *EVI1-eGFP* clone 24. Source data are provided as a Source Data file. **d** Flow cytometry plot presenting eGFP after *EVI1* knockdown (KD), compared to the control and scrambled shRNA in K562 *EVI1-eGFP* clone 24. **e** Schematic overview of the generation of a t(3;8)(q26;q24) in vitro using CRISPR-Cas9 technology, referred to in short as t(3;8). **f** Sorting strategy to enrich twice for cells with high *EVI1-eGFP* expression and select eGFP positive single clones with a t(3;8). **g** Flow cytometry plot presenting eGFP levels in t(3;8) K562 clones compared to the parental K562 *EVI1*-eGFP clones. Two parental clones (8 and 24), and four t(3;8) clones (8-3, 8-4, 24-7, and 24-12) are shown. **h** PCR amplicon covering the 3q26;8q24 breakpoint K562 *EVI1-eGFP* cells harboring a t(3;8), PCR for all single t(3;8) clones are provided in the Source Data file. **i** No significant difference in *MYC* expression (relative to *PBGD* expression) was observed between the K562 *EVI1-eGFP* parental clones (8 and 24) and the K562 *EVI1-eGFP* t(3;8) clones (8-3, 8-4, 24-7 and 24-12). Statistical test: ordinary one-way ANOVA (ns = not significant). The error bar represents the standard deviation (SD). **j** Significantly higher *EVI1* expression (relative to *PBGD* expression) shown by qPCR in the t(3;8) ($N = 4$) clones, compared to the parental clones ($N = 2$, $P = 0.0016$) and WT K562 ($P = 0.0051$). Statistical test: ordinary one-way ANOVA. The error bar represents the standard deviation (SD). **k** eGFP expression relative to *PBGD* shown by qPCR in the t(3;8) ($N = 4$) clones, compared to the parental clones ($N = 2$, $P < 0.0001$) and WT K562 ($P < 0.0001$). Statistical test: ordinary one-way ANOVA. The error bar represents the standard deviation (SD). **l** Western blot shows lower EVI1 levels for *EVI1* shRNA directed knockdown (KD), as compared to the control and scrambled shRNA in K562 *EVI1-eGFP* t(3;8) clone 24-7. Source data are provided as a Source Data file. **m** Flow cytometry plot presenting eGFP after *EVI1* shRNA directed knockdown (KD), as compared to the control and scrambled shRNA in K562 *EVI1-eGFP* t(3;8) clone 24-7.

4C-seq taking the *EVI1* promoter as a viewpoint revealed that the strong interaction with the *MYC* SE was severely diminished in the eGFP negative fraction upon deletion of module C (Fig. 4g). This loss of chromosomal interaction was also observed taking the *MYC* SE as a viewpoint (Fig. 4h).

Deletions of enhancer modules D and I affected neither *EVI1* expression nor enhancer-promoter looping (Fig. 4b and Supplementary Fig. 3d–e). Our data demonstrate that aberrant *EVI1* expression in t(3;8) AML depends on a single enhancer module within the *MYC* SE that recruits a cluster of key

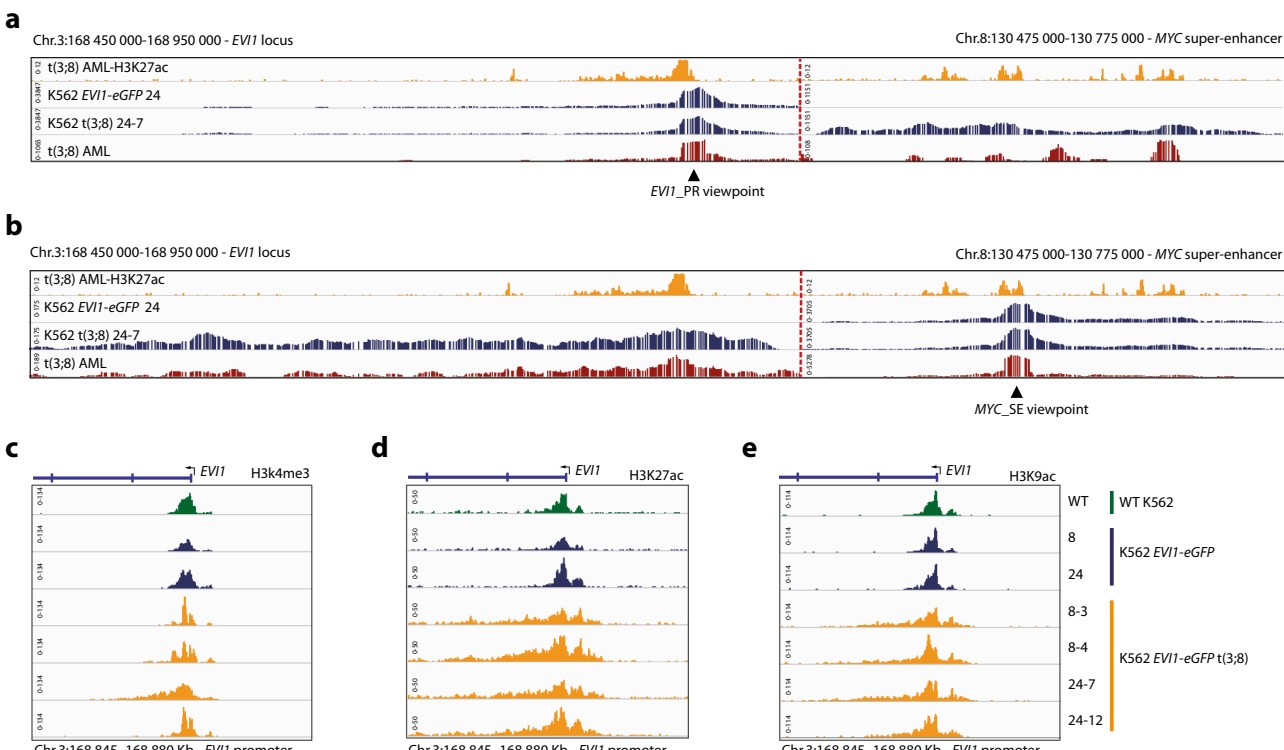

**Fig. 3 EVI1 promoter hyperactivation upon interaction with MYC SE in t(3;8) AML. a** Chromatin interaction shown by 4C-seq data, using the EVI1 promoter as the viewpoint (triangle symbol). The upper panel shows H3K27ac ChIP-seq data of a t(3;8) primary AML (AML-17). Indicated by H3K27ac signal peaks, on the left is the EVI1 promoter and on the right is the −1.7 Mb MYC super-enhancer, separated by a dotted red line. In the first 4C track (blue), parental K562 EVI1-eGFP clone 24; in the second, K562 t(3;8) EVI1-eGFP clone 24-7 (also blue); and in the bottom track (red), data of a primary t(3;8) AML (AML-17). **b** Similar to **a**, but using the MYC super-enhancer as the viewpoint (triangle symbol). The long stretch (500 Kb) of chromosomal interaction shown for the K562 t(3;8) EVI1-eGFP clone 24-7 shows a high resemblance with the interaction seen for the primary t(3;8) AML (AML-17) (second blue and red tracks, respectively). **c** H3K4me3 ChIP-seq data for K562 EVI1-eGFP parental lines (blue), the t(3;8) clones 8-3, 8-4, 24-7, 24-12 (orange), and K562 WT (green). A peak located on the EVI1 transcriptional start site marks the promoter region. **d** H3K27ac ChIP-seq data, comparing EVI1 promoter activation of the four t(3;8) clones (orange) to the EVI1-eGFP parental lines (blue) or K562 WT (green). **e** H3K9ac ChIP-seq data, confirming the hyperactivation of the EVI1 promoter in the t(3;8) clones (orange), compared to the parental lines (blue) or K562 WT (green).

hematopoietic transcription factors and facilitates promoter-enhancer looping.

**CTCF binding sites in MYC SE are involved in the interaction with the EVI1 promoter.** The EVI1 promoter interacts with the MYC SE over a long stretch of chromatin (275 Kb) with multiple zones of strong interaction indicative of a highly organized enhancer-promoter interaction (Fig. 5a). These high interaction zones in the MYC SE were associated with enhancer modules, but also with CTCF/cohesin binding based on ChIP-seq data (Fig. 5a). Notably, CTCF binding motifs in the MYC SE are arranged in a convergent orientation to that of a CTCF binding site upstream of the EVI1 promoter, suggesting the existence of a CTCF-facilitated enhancer-promoter loop. Using CRISPR-Cas9 technology, we sequentially deleted every CTCF binding site in the MYC SE. The deletions and their effect on EVI1 expression were shown by PCR and eGFP flow cytometry (Fig. 5b). A fraction of cells lost eGFP expression upon deletion of each of the CTCF binding sites in the MYC SE. The CTCF site closest to module C (CTCF2) was deleted and cells were sorted based on eGFP expression. A severe loss of promoter-enhancer interaction was observed in the eGFP negative cells (Fig. 5c and Supplementary Fig. 5a). This strongly supports a role for CTCF/cohesin in the promoter-enhancer complex formation and maintenance, and consequently in EVI1 regulation in t(3;8) AML.

**CTCF binding site upstream of the EVI1 promoter hijacks the MYC SE in t(3;8) AML.** Upstream of the EVI1 promoter a CTCF binding site in the forward orientation (CTCF EVI1_PR) was found by ChIP-seq and motif analysis (Figs. 5a and 6a). Deletion of this CTCF binding region caused loss of EVI1 expression as determined by eGFP flow cytometry. This loss of eGFP expression was comparable to the loss of expression upon deletion of the MYC SE CTCF sites (Fig. 5b). Deletion of this CTCF site also caused a severe loss of promoter-enhancer looping in eGFP negative cells, as measured by 4C-seq (Fig. 5d and Supplementary Fig. 5b). ChIP-seq showed that CTCF occupancy upstream of the EVI1 promoter was indeed reduced upon deletion of this site (Fig. 5e). CTCF occupancy at other CTCF binding sites, e.g., upstream of the MYC promoter (Fig. 5f), was not affected. Aiming to specifically target the CTCF binding and not other transcription factor binding motifs within this genomic region, more subtle mutations were made close to the CTCF binding motif using a single sgRNA (Fig. 6a). The mutations introduced by this single sgRNA strongly downregulated eGFP/EVI1 expression (Fig. 6b). A high mutation frequency was obtained in the eGFP negative sorted cells near the CTCF motif (Fig. 6c, d). These mutations led to a decrease of CTCF binding specifically at this site (Fig. 6e, f) and a severe loss of enhancer-promoter interaction (Fig. 6g) in the eGFP negative sorted cells. Taken together, these data demonstrate an important role for the CTCF binding site upstream of the EVI1 promoter in the hijacking of the MYC SE and the hyperactivation of EVI1.

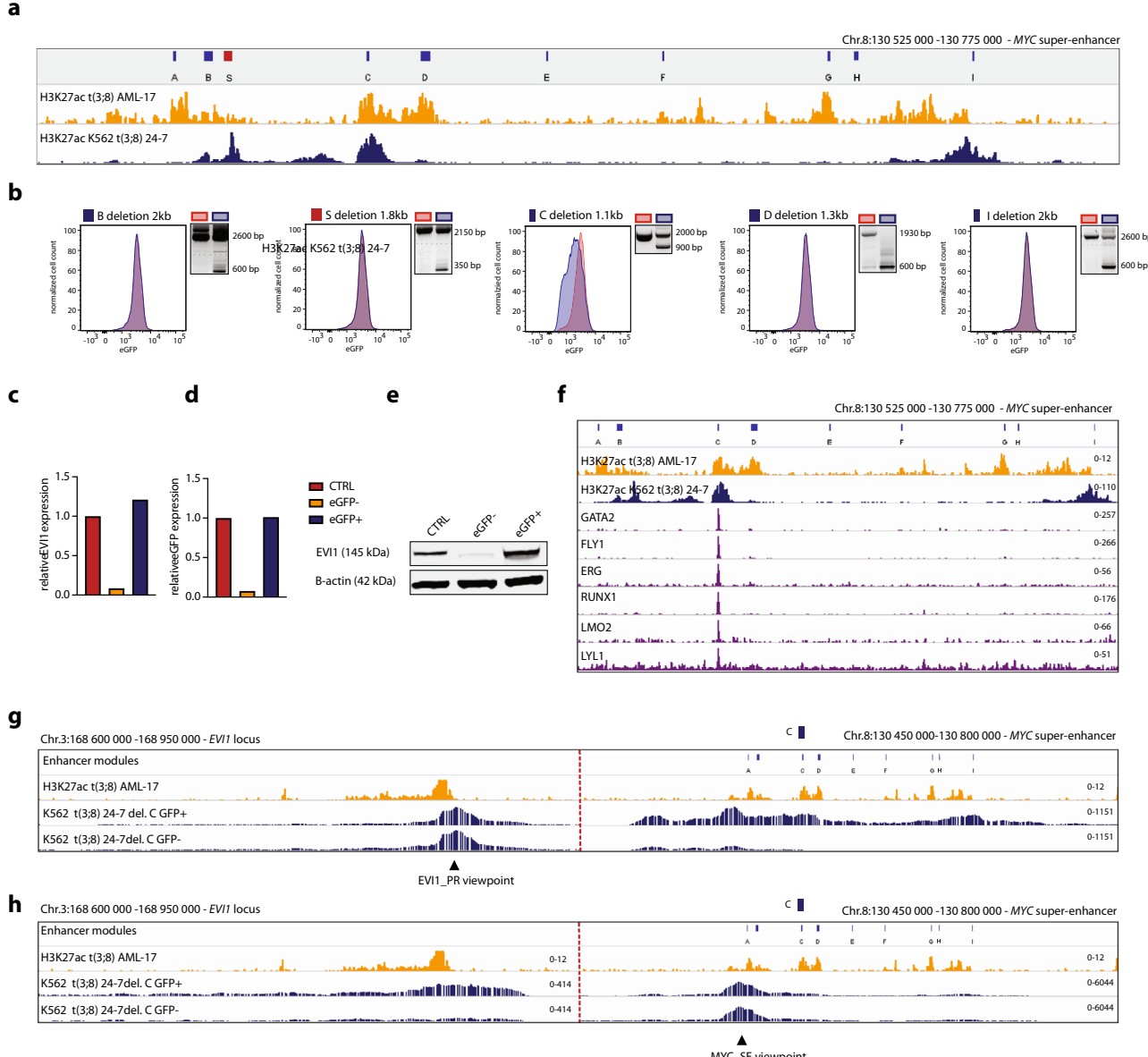

**Fig. 4 One critical enhancer module in the MYC SE drives EVI1 transcription. a** Overview of the *MYC* super-enhancer, with previously characterized individual enhancer modules A-I[28] and added module S based on high H3K27ac signal at this location in all K562 *EVI1-eGFP* t(3;8) clones. Underneath, H3K27ac of a primary t(3;8) AML (AML-17, orange) and of K562 t(3;8) *EVI1-eGFP* clone 24-7 (blue). **b** Flow cytometry plots (clone 24-7) shown for each indicated enhancer module deletion. In red are the control cells (no Cas9) and in blue are the cells carrying the deletion. On the right of each graph, the successful deletion of each element is shown by PCR. Source data are provided as a Source Data file. **c** *EVI1* expression relative to *PBGD* by qPCR in eGFP− and eGFP+ sorted cell fractions after deletion of enhancer module C. The bars represent only one data point, from the same experiment as the flow cytometry data shown in panel **b**. **d** *eGFP* expression relative to *PBGD* by qPCR in eGFP− and eGFP+ sorted cell fractions after deletion of enhancer module C. The bars represent only one data point, from the same experiment as the flow cytometry data shown in panel C. **e** EVI1 protein levels by Western blotting in eGFP− and eGFP+ sorted fractions after deletion of enhancer module C. Source data are provided as a Source Data file. **f** *MYC* SE element C recruits a set of HSPC-active transcription factors shown by ChIP-seq data of CD34 + cells (purple tracks[29]), H3K27ac of primary t(3;8) AML (AML-17, orange), and of a K562 t(3;8) (clone 24-7, dark blue) to illustrate enhancer modules. **g** Chromatin interaction shown by 4C-seq data, using the *EVI1* promoter as the viewpoint (triangle symbol). The *EVI1* promoter and the −1.7 Mb *MYC* SE are shown on the left and right sections, respectively, separated by a dotted red line. The upper orange panel shows H3K27ac ChIP-seq of a t(3;8) primary AML (AML-17). In blue, 4C-seq tracks of K562 *EVI1-eGFP* t(3;8) clone 24-7 cells in which the enhancer module C was deleted. In the upper blue track, eGFP+ sorted cells, and in the lower blue track, eGFP− cells. **h** Same as **g**, but using the *MYC* super-enhancer as a viewpoint (triangle symbol).

**CTCF enhancer-docking site upstream of the EVI1 promoter is preserved in all 3q26-rearranged AMLs.** The essential role of the CTCF binding site upstream of the *EVI1* promoter in mediating the interaction with a hijacked enhancer would predict that this site remains unaffected in 3q26-rearranged AMLs. Indeed, all breakpoints of t(3;8) AMLs analyzed were found upstream of this

CTCF site, placing the *MYC* SE 5' of *EVI1* (Figs. 1a and 7a). In t(3;3)(q21;q26) AML the *GATA2* enhancer similarly translocates 5' of the *EVI1* promoter and of the CTCF binding site (Fig. 7a)[15]. In AML with inv(3)(q21q26) the *GATA2* enhancer translocates 3' of *EVI1*[15] (Fig. 7a), leaving the enhancer-interacting CTCF site in position with respect to *EVI1* as well. We collected samples from

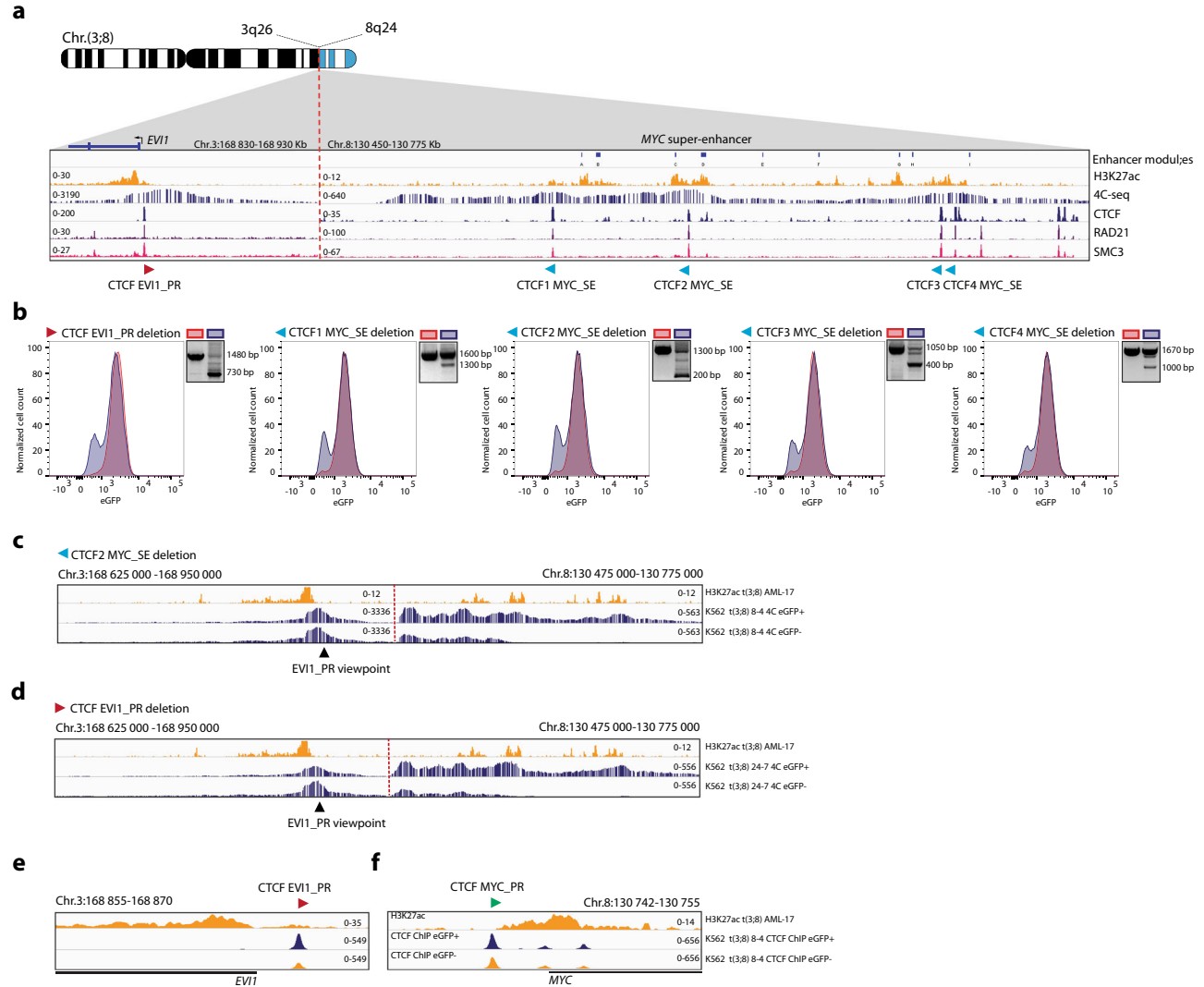

**Fig. 5 CTCF binding sites in *MYC* SE are involved in the interaction with the *EVI1* promoter. a** Schematic overview of the t(3;8) model, zoom-in on the breakpoint with 3q26 on the left and 8q24 on the right (separated by a dotted red line). The *EVI1* promoter and the *MYC* super-enhancer are illustrated by H3K27ac ChIP-seq data of primary t(3;8) AML (AML-17) in orange on top. Below in blue, 4C-seq data showing the interaction pattern between the *EVI1* promoter and the *MYC* enhancer modules in K562 t(3;8) clone 24-7. The lower 3 tracks show ChIP-seq data of CTCF (blue in K562 t(3;8) clone 24-7) and the cohesin subunits RAD21 (purple) and SMC3 (pink), both in K562 WT and retrieved from ENCODE[62]. **b** Flow cytometry plot of K562 *EVI1-eGFP* t(3;8) clone 8-4 cells after deletion of the indicated CTCF binding site (blue graph), and in red the control cells (no Cas9). On the right of the graph, the deletion is shown by PCR. Source data are provided as a Source Data file. **c** Chromatin interaction at the *MYC* SE in eGFP+ (upper blue track) and eGFP− cells (lower blue track), shown by 4C-seq with the *EVI1* promoter as viewpoint, after deletion of the *MYC* SE CTCF2 binding site. The top H3K27ac ChIP-seq track (orange) shows the presence of the active *EVI1* promoter and the modules of the *MYC* SE. **d** Chromatin interaction at the *MYC* SE in eGFP+ (upper blue track) and eGFP− cells (lower blue track), shown by 4C-seq with the *EVI1* promoter as viewpoint after deletion of the *EVI1* PR CTCF binding site (indicated by the red arrow, corresponding to the CTCF *EVI1_PR* locus in A). The top H3K27ac ChIP-seq track (orange) shows the presence of the active *EVI1* promoter and the modules of the *MYC* SE. **e** CTCF ChIP-seq presenting CTCF occupancy in eGFP− cells (clone 8-4, orange) compared to eGFP + cells (clone 8-4, blue) at the CTCF binding site upstream of the *EVI1* promoter after deletion of this CTCF *EVI1_PR* site. The top H3K27ac ChIP-seq track (orange) shows the presence of the active *EVI1* promoter and the modules of the *MYC* SE. **f** The same CTCF ChIP-seq tracks are shown (clone 8-4), but now presenting unchanged CTCF occupancy at the CTCF binding site upstream of the *MYC* promoter.

AML patients with translocations t(2;3)(p21;q26), t(3;7)(q26;q24) or t(3;6)(q26;q11) and carried out 3q-seq (Fig. 7a and Supplementary Fig. 6a). Irrespective of whether a translocation had occurred 3' or 5' of *EVI1*, the CTCF binding site flanking the *EVI1* promoter was never disrupted, suggesting a key role for this binding site in this AML subtype. Accordingly, ChIP-seq revealed constitutive binding of CTCF to this location across various leukemias, including not only 3q26-rearranged AMLs but also other AMLs and acute lymphoid leukemia (Supplementary Fig. 6b).

Enhancers of the genes *GATA2*[15] and *MYC* are, respectively, responsible for *EVI1* activation in inv(3)/t(3;3) and t(3;8) AML. Using 3q-seq, we observed that regions near the genes *CDK6* (6q11), *ARID1B* (7q24), and *THADA* (2p21) had been translocated to *EVI1* in t(3;6), t(3;7) or t(2;3) AML, respectively (Fig. 7b, c and Supplementary Fig. 6). All these genes are expressed in HSPCs[30]. Similar to the *MYC* SE in t(3;8) AML (Fig. 4f), we found strong regulatory regions close to these, illustrated by H3K27ac and hematopoietic transcription factor binding (Fig. 7b–d, Supplementary Fig. 6a). These commonalities suggest

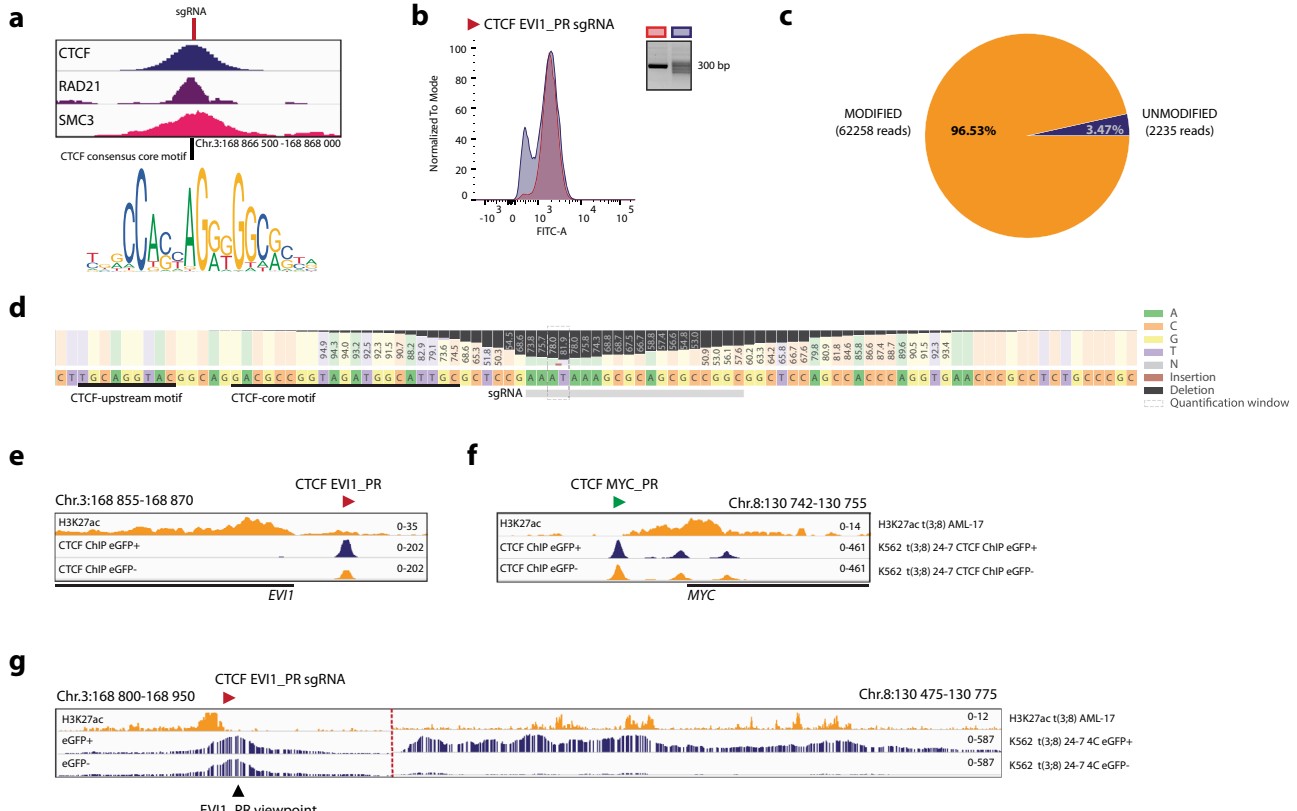

**Fig. 6 CTCF binding site upstream of the *EVI1* promoter hijacks a *MYC* SE in t(3;8) AML. a** ChIP-seq data of CTCF (blue) and the cohesin subunits RAD21 (purple) and SMC3 (pink) in K562, with a zoom-in on the *EVI1* promoter binding site. The vertical line indicates the exact cleavage site of the sgRNA and the CTCF motif as described by JASPAR[63] below. **b** Flow cytometry overlay plot after targeting the CTCF *EVI1*_PR binding site by sgRNA (clone 8-4, blue graph) and in red the control cells (clone 8-4, no Cas9). On the right of the graph, the mutations introduced by the single sgRNA in the amplicon over the cutting site are shown by PCR. Source data are provided as a Source Data file. **c** Amplicon-seq data showing the percentage of modified (orange) and unmodified (blue) reads in the eGFP− sorted cell fraction after targeting CTCF *EVI1*_PR. **d** Amplicon-seq data showing the mutations in the nucleotides around the Cas9 cleavage site, in the eGFP− sorted cell fraction after targeting CTCF *EVI1*_PR with sgRNA. The bars and numbers indicate the percentage of reads found with the particular mutation, below the locations of the sgRNA (gray bar) and the CTCF motifs (black lines). **e** CTCF ChIP-seq presenting CTCF occupancy in the eGFP+ (clone 24-7, blue), and eGFP− (clone 24-7, orange) fractions after targeting CTCF *EVI1*_PR with the sgRNA. The top H3K27ac track (orange) indicates the presence of an active promoter. **f** The same CTCF ChIP-seq tracks as in **e** are shown, but here presenting unchanged CTCF occupancy at the CTCF binding site upstream of the *MYC* promoter. **g** Chromatin interaction at the *MYC* SE for eGFP+ and eGFP− cells (clone 24-7), shown by 4C-seq with the *EVI1* promoter as viewpoint after targeting the CTCF motif with the sgRNA.

a shared mechanism for *EVI1* activation in all 3q26-rearranged leukemias, whereby an active hematopoietic enhancer is hijacked by a CTCF-mediated loop with the *EVI1* promoter. To validate this hypothesis, we targeted the *EVI1* CTCF binding site in MUTZ3-*EVI1*-eGFP, an inv(3) cell line engineered with eGFP as a reporter for *EVI1*[31]. In this model, mutations of the CTCF motif in a fraction of cells resulted in the loss of eGFP and EVI1 expression (Supplementary Fig. 7a–c), as well as loss of CTCF binding (Supplementary Fig. 7d). Altogether, these results confirm the role of this CTCF binding site in enhancer hijacking leading to *EVI1* overexpression.

## Discussion

In this study, we investigated how *EVI1* is deregulated in AML with a translocation t(3;8)(q26;q24). Using an *EVI1-eGFP* t(3;8) model, we demonstrated that hyperactivation of *EVI1* was driven by a hijacked *MYC* SE. One enhancer module within this *MYC* SE, previously reported as enhancer module C[28], was particularly essential for *EVI1* transcription. Module C is reported to be responsible for *MYC* expression in primary leukemic cells. The high accessibility of this module and the binding of a core set of hematopoietic transcription factors drive *MYC* expression in

HSPCs[28,29]. The other reported modules in the *MYC* SE, which did not affect *EVI1* transcription in a t(3;8) setting, may well be responsible for *MYC* transcription in other tissues[28]. Module C is the only element within the *MYC* SE to which early hematopoietic regulators bind, including GATA2, FLY1, ERG, RUNX1, LMO2, and LYL1[29]. Since those factors also bind to other enhancers that recurrently translocate to *EVI1* in t(2;3)(p21;q26), t(3;7)(q26;q24), t(3;6)(q26;q11), or inv(3)/t(3;3)(3q26;3q21) AML, we argue that *EVI1* expression is driven by a common mechanism. This is in line with our previous published data on a variety of atypical 3q26-rearranged AMLs[20]. The loci donating their enhancer to *EVI1* harbor genes that are normally expressed in early HSPCs, e.g., *MYC, ARID1B, CDK6, THADA,* or *GATA2.* Leukemias with high *EVI1* levels are chemotherapy-resistant and exhibit a unique gene expression signature comparable to that of CD34 + HSPCs[32]. This suggests that the cell of origin transformed in these leukemias is a very primitive hematopoietic progenitor cell.

The high-resolution 4C-seq data generated using our t(3;8) model revealed interaction of the *EVI1* promoter with the *MYC* SE, with multiple interaction zones associated with different enhancer modules indicative of a highly organized SE.

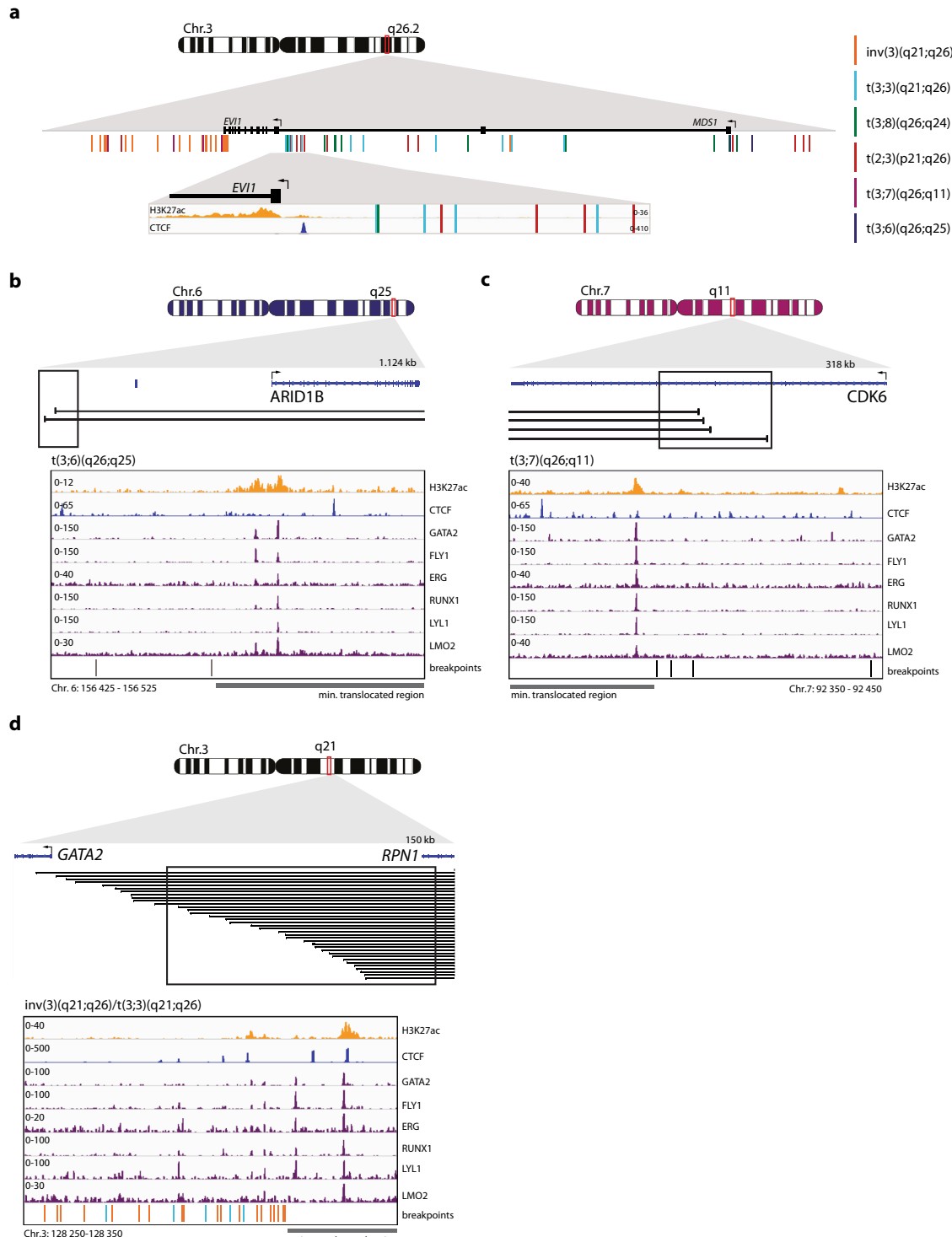

**Fig. 7 CTCF enhancer-docking site upstream of the *EVI1* promoter is preserved in 3q26-rearranged AML. a** Schematic depiction of Chr.3 with a zoom-in on the *EVI1* locus, indicating the exact breakpoints (detected by 3q-seq) of 3q26-rearranged AMLs as vertical lines. In the lowest zoom-in panel the *EVI1* promoter with a CTCF binding site upstream marked, respectively by H3K27ac (t(3;8) AML-17, orange) and CTCF (K562 t(3;8) clone 24-7, blue) ChIP-seq. **b** Schematic overview of Chr.6 and the locus where the breakpoints (black lines) were found by 3q-seq in t(3;6)(q26;q25) AML. The black box indicates the area of which the zoom-in is shown below. Zoom-in: putative enhancer indicated by H3K27ac (t(3;8) AML-17, orange), CTCF binding (K562 t(3;8) clone 24-7, blue) and HSPC active transcription factor recruitment (CD43+ cell[29], purple) at translocation site. The lines below indicate the exact breakpoints. The gray bar the minimal translocated region brought into close proximity of *EVI1* in that specific translocation. **c** Same as **b**, but here for t(3;7) (q26;q11) AMLs. **d** Same as **b**, but here for inv(3)/t(3;3)(q21;q26) AMLs. The exact translocated locus was previously shown to be an enhancer of *GATA2*[15].

Accordingly, Huang et al. defined the *MYC* SE as hierarchically organized. A hierarchical SE contains an enhancer module, referred to as hub enhancer, which is responsible for the structural organization of the SE and is distinctly associated with CTCF and cohesin binding[7]. Module C was characterized as a hub enhancer within the *MYC* SE in K562 cells[7]. Interestingly, the deletion of module C, while leaving CTCF binding sites intact, not only affected *EVI1* expression but also disrupted *MYC* SE-*EVI1* promoter interaction. Furthermore, mutations in the vicinity of the CTCF core binding region also resulted in the loss of interaction (Fig. 6d). Altogether, this suggests that transcription factors and co-activators occupying this location play a role in enhancer-promoter interaction, either independently or in cooperation with CTCF. Analogous to CTCF, YY1 contributes to DNA-looping, but preferentially occupies interacting enhancers and promoters[8]. Although there is no indication that YY1 binds directly to enhancer module C, we did find YY1 binding flanking this module (Supplementary Fig. 8). In embryonic stem cells (ESC), pluripotency factors e.g., OCT4, NANOG, and SOX2 recruit the mediator complex and stabilize the cohesin complex in order to facilitate cell-type-specific non-CTCF mediated enhancer-promoter looping[33]. In HSPCs a subunit of the mediator complex, MED12, co-localizes with key hematopoietic transcription factors, interacting with additional transcriptional co-activators to maintain enhancer activity[34]. We hypothesize that in t(3;8) and other 3q26-rearranged AMLs, enhancer-promoter interaction is facilitated by CTCF and cohesin, which is further stabilized by recruitment of co-factors by hematopoietic regulators (Supplementary Fig. 9).

All the CTCF binding motifs in the *MYC* SE are oriented in a 'reverse' fashion, allowing a CTCF/cohesin complex to be formed with the 'forward' CTCF binding site 2.6 kb upstream of the *EVI1* transcriptional start site (TSS). In all 3q26-rearranged AMLs, this upstream CTCF binding site was preserved with respect to *EVI1*. Interestingly, a CTCF binding site upstream of the *MYC* promoter has been reported to function as a docking site for enhancers driving *MYC* expression[6]. Our findings point to a very similar mechanism of transcriptional activation of *EVI1* in 3q26-rearranged AML. CTCF binding at this site proved to be absolutely critical for enhancer-promoter interaction and consequently indispensable for enhancer-driven *EVI1* transcription. Accordingly, it has been reported that promoters bound by CTCF, especially in enhancer deserts, are often dependent on long-range interactions[35].

Leukemias with 3q26 rearrangements depend on *EVI1*: interfering with EVI1 causes growth inhibition, differentiation, and ultimately death of leukemic cells[15,31]. Our data demonstrate mechanistic similarities between the distinct enhancer-driven *EVI1+* leukemias, suggesting that therapy for one subtype may be effective for all these AMLs. The *EVI1-eGFP* t(3;8) model is a valuable tool for compound screens to identify inhibitors of *EVI1* transcription that could constitute a promising treatment for these refractory leukemias. As enhancer-driven transcription is not limited to leukemia, this model can also be used to study (super-) enhancer biology and transcriptional regulation in a broader context.

## Methods
All the materials and resources used in this study are summarized in Supplementary Table 1. This study did not generate any unique codes. All software tools used in this study are freely or commercially available and listed in Supplementary Table 1. All primer names and sequences can be found in Supplementary Data 1, which is an Excel file with multiple tabs listing sgRNAs, qPCR primers, amplicon seq primers, PCR primers, and 4C primers.

**Patient material**. AML and T-ALL patient samples were collected either from the Erasmus MC Hematology department biobank (Rotterdam, The Netherlands) or from the MLL Munich Leukemia Laboratory biobank (Munich, Germany). Leukemic blast cells were purified from bone marrow or blood by standard diagnostic procedures. All patients provided written informed consent in accordance with the Declaration of Helsinki. The Medical Ethical Committee of the Erasmus MC has approved usage of the patient rest material for this study.

**Generation of *EVI1* expression cell model**. The plasmids to clone T2A-eGFP in frame with *EVI1* were designed and described by my colleagues as follows[31]. The repair template was generated using Gibson Assembly (NEB). Both homology arms were PCR amplified from MUTZ3 genomic DNA using Q5 polymerase (NEB). The first homology arm consists of a part of the intron and last exon of *EVI1* minus the STOP, the second homology arm consists of part of the 3'UTR with the PAM sequence of sgRNA omitted. The *T2A-eGFP* was PCR amplified from dCAS9-VP64_2A_GFP. All fragments were cloned using the Gibson assembly into the PUC19 backbone. The sgRNA sequence AGCCACGTATGACGTTATCA was cloned into pX330-U6-Chimeric_BB-CBh-hSpCas9. Cells were nucleofected with pX330 vector containing the sgRNA and Cas9 and the repair template using the NEON transfection system (Thermo Fisher Scientific) with buffer R and program 3 (1350 V, 10 ms, 4 pulses). GFP+ cells were sorted using a FACS AriaIII (BD Biosciences), and after two rounds of enrichment for cells expressing eGFP+, these cells were single-cell sorted and tested for proper integration. Subsequently, clones were named K562 *EVI1-eGFP*; multiple clones were obtained, but in this study, only clones 8 and 24 were used for further experiments.

**Generation of a t(3;8)(q24;q26) model**. K562 *EVI1-eGFP* clones (clone 8 and clone 24) were used as parental clones to generate the t(3;8)(q24;q26) clones. Based on the breakpoints (Chr.3:168.917.999-Chr.8:130.487.191) of the primary AML sample (#HF-80), sgRNAs were designed (using ChopChop V3[36], Supplementary Data 1) and mixed with purified Cas9 (IDT) to make ribonucleoproteins (RNPs). The NEON transfection system (Thermo Fisher Scientific) was used to get the RNPs into the K562 *EVI1-eGFP* clones. Three days after transfection the eGFP+ cells were sorted using the FACS AriaIII, and this enrichment process was repeated twice before eGFP+ single cells were sorted to produce single-cell clones. The clones were characterized for the designed specific t(3;8)(q24;q26) translocation by PCR (primers in Supplementary Data 1), Sanger-seq, flow cytometry, and FISH.

**Cytogenetics: karyotype and FISH**. Diagnostic cytogenetics for all samples was performed by each of the institutes mentioned above. For this study, samples were selected based on t(3;8)(q26;q24) rearrangements detected by karyotyping and/or *MECOM* interphase fluorescence in situ hybridization (FISH). FISH and classic metaphase karyotyping were performed and reported according to standard protocols based on the International System of Human Cytogenetics Nomenclature (ISCN) 2017[37]. For both patient samples and K562 clones *MECOM* FISH was performed according to the manufacturer's protocol, using the *MECOM* t(3;3); inv(3)(3q26) triple color probe (blue, green, red, Cytocell, LPH-036). For the characterization of the K562 *EVI1-eGFP* t(3;8) clones, the *MECOM* FISH was combined with: CEP8 (cen.8, blue), IGH (14q32, green), C-MYC(8q24, orange) (Vysis, 04N10-020), and C8 (Vysis, SpO, 07J22-008).

**Targeted chromosomal region 3q21.1-3q26.2 DNA sequencing (3q-seq)**. Genomic DNA was fragmented with the Covaris shearing device (Covaris), and sample libraries were constructed with the KAPA Hyper Prep Kit (Roche). After ligation of adapters and an amplification step, target sequences of chromosomal regions 3q21.1-q26.2 were captured by using custom in-solution oligonucleotide baits (Nimblegen SeqCap EZ Choice XL). Amplified captured sample libraries were paired-end sequenced (2 × 100 bp) on the HiSeq 2500 platform (Illumina) and aligned against the Human Genome Assembly 19 (hg19) using the Burrows-Wheeler aligner[38] v0.7.17. All chromosomal aberrations, such as translocations and inversions, were determined with BreakDancer v1.1[39].

**RNA isolation, quantitative PCR (qPCR), and RNA sequencing**. RNA was isolated using TRIzol (Invitrogen) or the Allprep DNA/RNA mini kit (Qiagen).
cDNA was synthesized using SuperScript II Reverse transcriptase (Invitrogen). Quantitative real-time RT-PCR was performed on the 7500 Fast Real-time PCR System (Thermo Fisher Scientific) with 10 μL Fast Sybr Green Master Mix (Thermo Fisher Scientific), 2 μL of cDNA, and primers listed in Supplementary Data 1. Relative levels of gene expression were calculated using the ΔΔ Ct method[40]. For qPCR data one-way ANOVA (GraphPad PRISM) was performed to indicate the level of significant differences between clones or conditions. For qPCR data of cells directly after FACS, no statistical test could be performed due to the limited number of cells (Fig. 4c, d). RNA-seq data from non-3q26 AMLs and CD34+ have been previously published in ref. [41] and are accessible at the European Genome-phenome Archive (EGA) under accession number EGAS00001004684.
Sample libraries were prepared using 500 ng of input RNA according to the KAPA RNA HyperPrep Kit with RiboErase (HMR) (Roche) using Unique Dual Index adapters (Integrated DNA Technologies, Inc.). Amplified sample libraries were paired-end sequenced (2 × 100 bp) on the Novaseq 6000 platform (Illumina). Salmon[42] v0.13.1 was used to quantify the expression of individual transcripts, which were subsequently aggregated to estimate gene-level abundances with

txtimport[43]. Differential gene expression analysis of count estimates from Salmon was performed with DEseq2[44]. The results of this analysis were depicted as a boxplot using the ggplot2 R package.

**Cell lines and culture**. K562 cells were cultured in RPMI 1640 + L-glutamine (Hyclone SH30027.LS), 10% fetal calf serum (FCS, Gibco), and 50 U/mL penicillin and 50 μg/mL streptomycin (Gibco 15140-163). Cells were incubated at 37 °C and 5% $CO_2$ and passaged every 3–4 days to 100,000 cells/ml. A previously generated MUTZ3 *EVI1*-eGFP cell line was cultured in αMEM (HyClone) with 20% fetal calf serum (FCS, Gibco) and 20% conditioned 5637 medium[31]. Unique biological materials are available upon request by contacting the corresponding author.

**Genome editing**. CRISPR-Cas9 technology was used to make mutations or deletions in the regions described in the results section. All primer sequences to generate sgRNAs can be found in Supplementary Data 1 and were ordered from IDT. By in vitro transcription, sgRNAs were produced as described above for the generation of the t(3;8). In short: the constant and specific oligos were annealed and filled in 20 min 12 °C by T4 polymerase (NEB, M0203S), sgRNAs were produced by in vitro transcription using HIScribe T7 High-Yield RNA Synthesis kit (NEB, E2050S) 3–4 h, 37 °C, DNA was eliminated by Turbo DNase (Thermo Fisher Scientific, AM2238), 15 min, 37 °C. The sgRNAs were concentrated and purified using RNA clean and concentrator -25 (Zymo Research, R1017). The concentration of sgRNAs was estimated using Qubit RNA BR assay (Invitrogen, Q10210). Ribonucleoproteins (RNPs) were made by mixing purified Cas9 protein (IDT, Nucleofection of all K562 clones was done with NEON transfection buffer T (Thermo Fisher Scientific) and settings 1350 V, 10 ms, 4 pulses. Nucleofection of MUTZ3 *EVI1*-eGFP cells was done with NEON transfection buffer R (Thermo Fisher Scientific) and settings 1500 V, 20 ms, 1 pulse. After a minimum of 72 h post nucleofection DNA or RNA was extracted (DNA Quick extract, Epicenter or Qiagen Allprep DNA/RNA mini, #80204) or cells were harvested for further analysis by, respectively, PCR, qPCR, or flow cytometry analysis/FACS sorting.

The targeted CTCF motifs in the *EVI1* promoter were identified using the CTCFBSDB 2.0 database[45]. CTCF motif orientation at the *EVI1* promoter and the *MYC* SE was retrieved from the JASPAR database (release 2020)[46].

**Flow cytometry and sorting (FACS)**. Flow cytometric analysis or cell sorting was performed using the FACS Canto or the FACS Aria flow cytometer (BD Biosciences). Cells were gated on viability and single cells using FSC/SSC, eGFP intensity levels were measured using the FITC channel. Data were analyzed using FACS Diva v9.0 and FlowJo v10.0.

**PCR and primers**. For all PCRs used to detect translocations, point mutations, or deletions; Q5 High-Fidelity DNA polymerase was used following the manufacture's protocol (NEB, #0491) and primers listed in Supplementary Data 1. PCR products were purified using a Qiaquick PCR purification kit. Purified PCR products were subjected to Sanger sequencing on an Applied Biosystems 3730 device using a BigDye™ Terminator v1.1 Cycle Sequencing Kit (Thermo Fisher Scientific) and primers listed in Supplementary Data 1.

**Amplicon sequencing**. To check mutations after targeting with CRISPR-Cas9 we performed amplicon sequencing using the Illumina PCR-based custom amplicon sequencing method using the TruSeq Custom Amplicon index kit (Illumina). The first PCR was performed using Q5 polymerase (NEB), the second nested PCR was with KAPA HiFi HotStart Ready-mix (Roche). Samples were sequenced paired-end (2× 250 bp) on a MiSeq (Illumina). Reads were trimmed with trimgalore[47] v0.4.4 to remove low-quality bases and adapters and subsequently aligned to the human reference genome build hg19 with BBMap[48] v34.92 allowing for 1000 bp indels. Mutations introduced by genome editing were analyzed and visualized using CRISPResso[49] v2.0.27.

**Western blotting**. Cells were lysed using the NE-PER Nuclear and Cytoplasmic Extraction Kit (Thermo Fisher Scientific) following the manufacturer's protocol and the nuclear extract was used for Western Blotting of EVI1 (#2265 Cell Signaling, dilution: 1:1000). As a loading control, an antibody against B-Actin (clone AC15, A5441, Sigma, dilution: 1:10,000) was used. The Odyssey infrared imaging system (Li-Cor) was used for the visualization of the protein levels.

**4C sequencing**. Chromosome Conformation Capture Sequencing (4C-seq) sample preparation was performed using 10 million cells[50]. In short, genomic regions that are spatially proximal in the cell nucleus were fixated by formaldehyde-induced crosslinks. The DNA was fragmented with DpnII as a primary restriction enzyme, Csp6I as a secondary 4 bp-cutter. To identify and quantify fragments that were ligated to the genomic region of interest, a two-step PCR was performed[51]. The first PCR step was an inverse PCR with viewpoint-specific primers that are listed in Supplementary Data 1. In the second PCR step, universal primers were used that contain the Illumina adapters. The amplicons were subjected to next-generation sequencing on the IIlumina NovaSeq 6000 platform.

Demultiplexing and clipping of the primer sequences were performed by an in-house algorithm. Subsequently, the reads of each viewpoint were aligned against the human genome (hg19) with bowtie[52] v1.1.1. Reads not mapping to fragments determined by the restriction site positions of the chosen primary and secondary restriction enzymes were removed by an in-house algorithm. Generated BAM files were transformed into WIG files with an in-house tool, applying a running mean (window size 21) for signal smoothing of peaks. The data were also normalized to reads per million (RPM). In all figures, the tracks were displayed on the Integrative Genomics Viewer (IGV)[53] v2.8 using "group auto-scale" to compare relevant samples.

**ChIP sequencing**. Chromatin immunoprecipitation sequencing (ChIP-seq) experiments were performed using 10 to 20 million cells. Cells were cross-linked with 1% formaldehyde. Chromatin was isolated using lysis buffer A (50 mM Tris pH 8, 10 mM EDTA, 1% SDS). In the CTCF ChIP, 0.5% EPIGEN BB was added to the lysis buffer A. In the RUNX1 ChIP at least 30 million cells were double crosslinked with 2 mM disuccinimidyl glutarate followed by 1% formaldehyde. Chromatin of double crosslinked cells was isolated using lysis buffer B (10 mM Tris pH 7.5, 74 mM NaCl, 3 mM MgCl2, 1 mM CaCl2, 4% NP40, 0.32% SDS). The chromatin was sonicated with a Bioruptor device (Diagenode) using the following settings: 10 cycles of 30 s on, 30 s off.

Immunoprecipitation of cross-linked chromatin was performed with antibodies directed against H3K27Ac (Diagenode C15410196, 2.5 μg), H3K9Ac (Diagenode C15410004, 2.5 μg), H3K4me3 (Diagenode C15410003, 2.5 μg), RUNX1 (Abcam Ab23980, 5 μg) in IP dilution buffer (16.7 mM Tris pH 8, 1.2 mM EDTA, 167 mM NaCl, 1.1% Triton, 0.01% SDS) or CTCF (Cell Signaling 2899S, 5 μg) in CTCF IP dilution buffer (20 mM Tris, 2 mM EDTA, 100 mM NaCl, 0.5% Triton). Chromatin bound antibody was precipitated with prot G Dynabeads (Thermo Fisher Scientific) and washed with low salt buffer (20 mM Tris pH 8, 2 mM EDTA, 1% Triton, 150 mM NaCl), high salt buffer (20 mM Tris pH 8, 2 mM EDTA, 1% Triton, 500 mM NaCl), LiCl buffer (10 mM Tris, 1 mM EDTA, 0.25 mM LiCl, 0.5% IGEPAL, 0.5% Sodium-Deoxycholate) and TE (10 mM Tris pH 8, 1 mM EDTA). Chromatin was eluted in elutionbuffer A (25 mM Tris, 10 mM EDTA, 0.5% SDS). In the CTCF ChIP and RUNX1 ChIP chromatin was eluted in elution buffer B (0.1 M Sodiumhydrogencarbonate, 1% SDS).

Crosslinks were reversed overnight at 65 °C in the presence of proteinase K (New England Biolabs). De-crosslinked material was purified using a QIAGEN PCR Purification Kit. The purified DNA was processed according to the Nextflex ChIP Sample Preparation Protocol (Perkin Elmer) or the Microplex library preparation kit V2 (Diagnode C05010013) and sequenced on the Illumina NovaSeq6000 platform. ChIP-seq reads were aligned to the human reference genome build hg19 with bowtie[52] v1.1.1 and bigwig files were generated for visualization with the bamCoverage tool from deepTools[54] v3.4.3, with the options–normalizeUsing RPKM --smoothLength 100 --binSize 20. Peak calling was performed with MACS2[55] v2.2.7.1 using default settings. Publicly available ChIP-seq data were downloaded from the Gene Expression Omnibus: RAD21 and SMC3 tracks in K562 cells generated by the ENCODE consortium[56], MED12 data in K562 published by the Aifantis group[34], and hematopoietic transcription factors in CD34+ cells generated by the Pimanda group[29]. In all figures displaying ChIP-seq data the y-axis shows normalized RPKM, and "group auto-scale" was used on IGV[53] v2.8 to compare relevant samples.

**ATAC sequencing**. Cells were washed using PBS and counted in a Bürker-Türk counting chamber. 50.000–100.000 cells were pelleted and resuspended in 1 mL ATAC lysing buffer containing: 0.3 M Sucrose, 10 mM Tris HCL pH 7.5, 60 mM KCl, 15 mM NaCl, 5 mM MgCl2, 0.1 mM EGTA, 0.1% NP40, 0.15 mM Spermine, 0.5 mM Spermidine and 2 mM 6AA. All components were derived from Sigma Aldrich. Cells were incubated in a lysis buffer for 3 min on ice. Cells were pelleted at 500 × g for 10 min at 4 °C and the supernatant was removed. Pelleted cells were resuspended in a 50 μL transposase mixture containing 25 μl 2× TD buffer, 2.5 μL TD1 transposase, 22.5 μl nuclease-free water (kit Illumina cat no 20034197). Samples were incubated for 30 min at 37 °C while mixing at 500 RPM in a heat block. Samples were immediately purified using the Qiagen min elute PCR purification kit following manufacturers' protocol. Transposase fragmented DNA was eluted in a 10 μL elution buffer. All DNA was used in a 4 cycle PCR amplification using Nextera i7-index and i5-index primers (Illumina). Five microliters of the 4 cycles of amplified material were used in taqman. ¼ of the maximum signal was determined and cycles were added to the remaining 45 μL library to avoid over-amplification of the ATAC library. Amplified libraries were purified using Agencourt AMPure XP beads using a 1;1,8 ratio. DNA was eluted using 30 μL EB buffer. Libraries were quantified using Qubit and PCR NEBnext library quant kit for Illumina (NEB). Size distribution was determined by running 1 ng library on a DNA-high sensitive chip (Agilent/Bioanalyzer).

ATAC-seq samples were sequenced paired-end 2 × 50 bp or 2 × 100 bp on the Hiseq 2500 and the Novaseq 6000 platforms (both Illumina). They were aligned against the human genome (hg19) with bowtie2[57] v2.3.4.1, allowing for a maximum 2000 bp insert size. Mitochondrial reads and fragments with mapping quality below 10 were removed. bigwig files were generated for visualization with the bamCoverage tool from deepTools v3.4.3[54], with the options --normalizeUsing RPKM--smoothLength 100--binSize 20. In all figures displaying ATAC-seq data

the y-axis shows normalized RPKM, and "group auto-scale" was used on IGV[53] v2.8 to compare relevant samples.

**Comparative analysis of modules in the MYC super-enhancer**. Quantification of H3K27ac and ATAC-seq reads was conducted in the different enhancer modules within the MYC super-enhancer, as defined in[28]. A BED file containing these modules was converted into GTF with the UCSC tools bedToGenePred and genePredToGtf[58]. Read counts in enhancer regions were computed with featureCounts[59] and differential analysis was conducted with the DESeq2 R package[44]. The results of this analysis were depicted as a boxplot using the ggplot2 package in R.

**SNP array**. DNA was isolated from K562 cells using the AllPrep DNA/RNA mini kit (Qiagen, #80204). All SNP arrays were performed at the Erasmus MC Department of Clinical Genetics (Rotterdam, The Netherlands) and analyzed as previously described[20,60,61]. In short, 200 ng of DNA was used as an input for a single array. DNA amplification, tagging, and hybridization were performed according to the manufacturer's protocol. The array slides were scanned on an iScan Reader (Illumina). Data analysis was performed using GenomeStudio version 2.0, KaryoStudio version 1.4 (Illumina, standard settings), and Nexus Copy Number 9.0 (BioDiscovery, El Segundo, CA, USA).

**Statistics and reproducibility**. The *EVI1* knockdown experiment shown in Fig. 2c and Supplementary Fig. 1d were performed in 2 biological replicas; in clone 8 and clone 24. The *EVI1* knockdown experiment shown in Fig. 2l and Supplementary Fig. 2f was performed in 3 biological replicas; in clone 8-4 and clone 24-7 and clone 24-12. The PCR over the t(3;8) breakpoint to identify single clones that harbored the translocation, shown in Fig. 2h, was done on over 20 single clones/biological replicas. Uncropped PCR gel pictures are provided in the Source Data file. The deletions induced in the *MYC* super-enhancer as shown by PCR in Fig. 4b (right panel) were performed in 3 biological replicas of which 2 t(3;8) clones: clone 8-4 and 24-7 and one control clone harboring a 3q/*MECOM* amplification: clone 24-2 (as characterized in Supplementary Fig. 4). The CTCF binding site deletions induced in the *MYC* SE as shown by PCR in Fig. 5b (right panel) and the single cut by shRNA#1 were performed as a minimum with 2 biological replicas in t(3;8) clone 8-4 and the control clone 24-2. However, most important experiments like deletion of enhancer module C, the deletion CTCF2 in the *MYC* SE of CTCF near the *EVI1* promoter of the single cut in the CTCF site at the *EVI1* promoter (sgRNA#1) were performed in 3 biological replicas in clone 8-4, 24-7 and control clone 24-2 and at least twice in the two separate t(3;8) clones. Uncut PCR gel pictures are provided in the Source data file. The western blot showing the EVI1 protein levels after sorting in Fig. 4e was performed in 2 biological replicas. Uncut blot pictures are provided in the Source data file. Genotyping of the K562 *EVI1-eGFP* clones (Supplementary Fig. 1b and 1c) was done for 10 single clones. Clones with the correct genotype were selected based on at least 3 different PCR methods. Uncut PCR gel pictures are provided in the Source data file. The FISH experiments were done by the Erasmus MC diagnostic lab following their verified experimental setup, the FISH experiments as shown in Supplementary Fig. 2c, 2d, 2e, and 4d were done on 4 separate clones each. All with similar results as shown in the pictures in the main manuscript. The PCR on the sorted fractions was performed in 3 biological replicas in 2 separate t(3;8) clones (1× clone 8-4 and 2 × 24-7). Uncut PCR gel pictures are provided in the Source data file. In Supplementary Fig. 4 the control clone 24-4 harboring the 3q/*MECOM* amplification is characterized. In total, we generated 4 clones (4 biological replicas) like this of which two clones are shown (24-1 and 24-2) in Supplementary Fig. 4b. The uncut PCR gel picture showing all 4 clones is provided in the Source data file. The deletions induced by CRISPR-Cas9 as shown in Supplementary Fig. 4g are done all at least in 2 biological replicas. This control clone 24-2 was always taken along in CRISPR-CAs9 experiments as a (negative) control for an effect on EVI1/eGFP expression.

**Reporting summary**. Further information on research design is available in the Nature Research Reporting Summary linked to this article.

## Data availability

The data that support this study are available from the corresponding author upon reasonable request. The ChIP-seq, 3q-seq, 4C-seq, and RNA-seq data derived from human patients generated in this study are available at the European Genome-phenome Archive (EGA), under the accession code EGAS00001004808. These data are available under restricted access due to data privacy laws, access can be obtained by contacting the Data Access Committee and signing a Data Access Agreement. Data derived from K562 have been uploaded to the ArrayExpress database under the following accession codes: E-MTAB-9958 (4C-seq), E-MTAB-9965 (ChIP-seq), E-MTAB-10785 (ATAC-seq), and E-MTAB-9937 (Amplicon-sequencing following CRISPR-Cas9 treatment). This study also used publicly available sequencing datasets. The 3q-seq data of the inv(3)/t(3;3) cell lines MUTZ3 and MOLM1 were downloaded from ArrayExpress, under the accession code E-MTAB-2224. The ChIP-seq data of a heptad of transcription factors in CD34+ cells generated by the Pimanda group[29] were downloaded from the Gene Expression Omnibus (GEO), under the accession code GSE38865. The ChIP-seq data of RAD21, SMC3, and YY1 generated by the

ENCODE consortium[56] were also downloaded from GEO, under the accession code GSE31477. The RNA-seq data of HSPCs generated by the Blueprint consortium[30] was accessed via the Blueprint Data Analysis Portal (http://blueprint-data.bsc.es/release_2016-08/). RNA-seq data from non-3q26 AMLs and CD34+ have been previously published in ref. [41] and are accessible at the EGA under accession number EGAS00001004684. Source data is provided with this paper. All uncut blots and gel pictures can be found in the Excel Source file. Source data are provided with this paper.

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

## Acknowledgements

The authors are indebted to their colleagues from the bone marrow transplantation group and the molecular and cytogenetics diagnostics laboratories of the Department of Hematology and Clinical Genetics at Erasmus University Medical Center for storage of samples and molecular analysis of the leukemia cells (M. Wattel, R. van der Helm, and P.J.M. Valk). For providing patient material, the authors are thankful to the MLL Münchner Leukämielabor GmbH in Germany. They also thank P. Sonneveld and their colleagues of the Hematology Department, especially those involved in FACS sorting (C. van Dijk), Next Generation Sequencing operating bioinformatics (R. Hoogenboezem), and all others for their input or expertise. We also thank N.J. Galjart and R. Stadhouders of the department of Cell Biology and Pulmonary medicine at the Erasmus University Medical Center for their input and expertise. This work was funded by grants and fellowships from the Dutch Cancer Society (R.D., R.M.-L., S.O., L.S.), Skyline DX (S.O.), the Daniel den Hoed, Erasmus MC Foundation (L.S.).

## Author contributions

S.O., L.S. and R.D. conceived of and designed the study. B.W., C.H. and T.H. provided study materials or patient samples. H.B.B. supervised all cytogenetic and FISH characterization. C.E-V., S.v.H., M.H., A.A.V., L.S., S.G. and S.O. performed experiments, data analysis, and interpretation. M.V. operated the FACS sorter for a significant part of the experiments. E.B. organized all NGS and was involved in data interpretation. R.M.-L. was responsible for all bioinformatics data processing and analysis of this study. S.O., R.M.-L., L.S. and R.D. wrote the manuscript.

## Competing interests

T.H. and C.H. are employees of and have equity ownership in MLL Munich Leukemia Laboratory. The remaining authors declare no competing interests.
