## [Peer Review File · Nature Communications]

The leukemic oncogene EVI1 hijacks a MYC super-enhancer by CTCF-facilitated loopsEditorial Note: This manuscript has been previously reviewed at another journal that is not operating a transparent peer review scheme. This document only contains reviewer comments and rebuttal letters for versions considered at *Nature Communications*.

REVIEWERS' COMMENTS

Reviewer #2 (Remarks to the Author):

The authors have constructively addressed my questions. The role of CTCF sites in the MYC enhancer region remains to be established as no effective gRNAs effectively target these sequences. The authors should review the title and abstract which currently overemphasizes the role of CTCF and CTCF sites in the MYC enhancer in looping/enhancer hijacking. The observation that a region different from CTCF is required for looping is most intriguing, yet the mechanism at play there remains to be established. In all this manuscript contains a large amount of data and represents a significant effort but in its present form adds little to the field and leaves open the most interesting questions.

RESPONSE TO REVIEWERS

Reviewer #2 (Remarks to the Author):

The authors have constructively addressed my questions. The role of CTCF sites in the MYC enhancer region remains to be established as no effective gRNAs effectively target these sequences. The authors should review the title and abstract which currently overemphasizes the role of CTCF and CTCF sites in the MYC enhancer in looping/enhancer hijacking. The observation that a region different from CTCF is required for looping is most intriguing, yet the mechanism at play there remains to be established. In all this manuscript contains a large amount of data and represents a significant effort but in its present form adds little to the field and leaves open the most interesting questions.

We appreciate that the reviewer is satisfied with our previous response to his questions. Although we did not specifically target the CTCF motifs in the MYC super-enhancer (SE) like we did for the CTCF binding site near *EVI1*, our findings provide sufficient evidence to infer an involvement of CTCF in the enhancer-promoter interaction:

- First, deletion of any CTCF-containing region of the MYC SE in t(3;8) K562 led to a marked decrease in GFP/*EVI1* expression (Figure 5A-B). Note that while deletion of module C (in close proximity to the CTCF2 site) also yielded a reduction of *EVI1* expression, deletions of modules B, S, D and I had no effect (Figures 4A-B). This argues for a specific role of CTCF-binding sites, i.e. the changes in *EVI1* expression are not simply the result of deleting an arbitrary section of the MYC SE.
- Second, the CTCF binding site near the *EVI1* promoter is in convergent orientation with the CTCF binding sites in the MYC SE, and 4C-seq shows particularly strong interaction peaks between the *EVI1* CTCF site and the MYC SE CTCF sites (Figure 5A). This suggests the existence of CTCF-facilitated loops between the *EVI1* promoter and the MYC SE.
- Third, the loss of *EVI1* upon deletion of the CTCF2 site was accompanied by a loss of interaction between the *EVI1* promoter and the MYC SE (Figure 5c, Supplementary Fig. 5a). Given that CTCF can be involved in enhancer-promoter loop formation, it stands to reason that the simultaneous loss of *EVI1* expression and chromatin interaction is due to loss of CTCF binding rather than other transcription factors in the deleted regions.
- Fourth, similar effects were observed upon deletion of the CTCF binding site near *EVI1* (Figure 5d, Supplementary Fig. 5b) and specific targeting of that CTCF binding motif (Figure 6g, Supplementary Fig. 5c). Since the *EVI1* CTCF site interacts with the convergent CTCF binding sites of the MYC SE in untreated t(3;8) K562, these findings can be explained by an involvement of MYC SE CTCF sites in loop formation.
- Fifth, the loss of *EVI1* expression and of MYC SE/*EVI1* interaction upon mutating the CTCF binding site near the *EVI1* promoter was accompanied by significant loss of CTCF binding, as measured by ChIP-seq (Figure 6e). This is in line with our conclusion that “*EVI1* hijacks the MYC super-enhancer by CTCF-facilitated loops”.

Taken together, our results support the notion that the CTCF sites near the promoter of *EVI1* and within the *MYC* SE are involved in loop formation and are indispensable for *EVI1* expression. However, as pointed out by the reviewer, we also show that other factors aside from CTCF are involved in the enhancer-promoter loop. Thus, we have tried to downplay the emphasis on CTCF in the title and the abstract, while at the same time conveying the message that CTCF is important in this context. The amended versions are presented below:

The leukemic oncogene *EVI1* hijacks a *MYC* super-enhancer by CTCF-facilitated loops

Chromosomal rearrangements are a frequent cause of oncogene deregulation in human malignancies. Overexpression of *EVI1* is found in a subgroup of acute myeloid leukemia (AML) with 3q26 chromosomal rearrangements, which is often therapy resistant. In AMLs harboring a t(3;8)(q26;q24), we observed the translocation of a *MYC* super-enhancer (*MYC* SE) to the *EVI1* locus. We generated an *in vitro* model mimicking a patient-based t(3;8)(q26;q24) using CRISPR-Cas9 technology and demonstrated hyperactivation of *EVI1* by the hijacked *MYC* SE. This *MYC* SE contains multiple enhancer modules, of which only one recruits transcription factors active in early hematopoiesis. This enhancer module is critical for *EVI1* overexpression as well as enhancer-promoter interaction. Multiple CTCF binding regions in the *MYC* SE facilitate this enhancer-promoter interaction, which also involves a CTCF binding site upstream of the *EVI1* promoter. We hypothesize that this CTCF site acts as an enhancer-docking site in t(3;8) AML. Genomic analyses of other 3q26-rearranged AML patient cells point to a common mechanism by which *EVI1* uses this docking site to hijack enhancers active in early hematopoiesis.